# Advances in Cryopreservation Strategies for 3D Biofabricated Constructs: From Hydrogels to Bioprinted Tissues

**DOI:** 10.3390/ijms26146908

**Published:** 2025-07-18

**Authors:** Kaoutar Ziani, Laura Saenz-del-Burgo, Jose Luis Pedraz, Jesús Ciriza

**Affiliations:** 1NanoBioCel Group, Department of Pharmacy and Food Sciences, Faculty of Pharmacy, University of the Basque Country UPV/EHU, Paseo de la Universidad 7, 01006 Vitoria-Gasteiz, Spain; kziani001@ikasle.ehu.eus (K.Z.); laura.saenzdelburgo@ehu.eus (L.S.-d.-B.); 2Biomedical Research Networking Center in Bioengineering, Biomaterials, and Nanomedicine (CIBER-BBN), 28029 Madrid, Spain; 3Bioaraba, NanoBioCel Research Group, 01002 Vitoria-Gasteiz, Spain; 4Tissue Microenvironment (TME) Laboratory, Aragón Institute of Engineering Research (I3A), University of Zaragoza, Campus Rio Ebro, Mariano Esquillor s/n, Edificio I+D+I, 50018 Zaragoza, Spain; 5Tissue Microenvironment (TME) Laboratory, Institute for Health Research Aragón (IIS Aragón), 50009 Zaragoza, Spain

**Keywords:** cryopreservation, 3D biofabrication, hydrogels, DMSO-free, tissue engineering

## Abstract

The cryopreservation of three-dimensional (3D) biofabricated constructs is a key enabler for their clinical application in regenerative medicine. Unlike two-dimensional (2D) cultures, 3D systems such as encapsulated cell spheroids, molded hydrogels, and bioprinted tissues present specific challenges related to cryoprotectant (CPA) diffusion, thermal gradients, and ice formation during freezing and thawing. This review examines the current strategies for preserving 3D constructs, focusing on the role of biomaterials as cryoprotective matrices. Natural polymers (e.g., hyaluronic acid, alginate, chitosan), protein-based scaffolds (e.g., silk fibroin, sericin), and synthetic polymers (e.g., polyethylene glycol (PEG), polyvinyl alcohol (PVA)) are evaluated for their ability to support cell viability, structural integrity, and CPA transport. Special attention is given to cryoprotectant systems that are free of dimethyl sulfoxide (DMSO), and to the influence of hydrogel architecture on freezing outcomes. We have compared the efficacy and limitations of slow freezing and vitrification protocols and review innovative approaches such as temperature-controlled cryoprinting, nano-warming, and hybrid scaffolds with improved cryocompatibility. Additionally, we address the regulatory and manufacturing challenges associated with developing Good Manufacturing Practice (GMP)-compliant cryopreservation workflows. Overall, this review provides an integrated perspective on material-based strategies for 3D cryopreservation and identifies future directions to enable the long-term storage and clinical translation of engineered tissues.

## 1. Introduction

The cryopreservation of biofabricated 3D constructs is an emerging and indispensable component of regenerative medicine and clinical tissue engineering. As the field advances toward the development of complex tissue equivalents, including vascularized grafts, organoids, and 3D-bioprinted scaffolds, there is a growing demand for cryopreservation strategies that can preserve not only cellular viability but also the architectural and functional integrity of constructs through the freeze–thaw cycle [1,2]. The applications include off-the-shelf tissue implants, patient-specific models for personalized medicine, and advanced therapeutic products such as bioprinted corneal epithelium and osteochondral grafts [3,4,5]. Other clinically oriented examples include cryopreserved hepatic microtissues used in drug toxicity testing and liver disease modeling, further emphasizing the need for cryopreservation protocols that ensure post thaw structural and functional integrity [6].

Despite this potential, current cryopreservation protocols face significant limitations. DMSO, the most commonly used cryoprotectant, is associated with cytotoxicity, osmotic stress, and residual toxicity that complicate its use in clinical-grade products [7,8]. Moreover, the diffusion of CPAs, ice formation dynamics, and thawing profiles are more complex in 3D constructs than in single-cell suspensions, making standard protocols inadequate [1,9,10]. These challenges are further compounded by the diversity of materials used in biofabrication—ranging from polysaccharides and proteins to synthetic polymers—each influencing cryostability in distinct ways [11,12].

Recent studies have proposed a shift toward DMSO-free or low-toxicity cryopreservation strategies by using biomaterials that offer both structural support and intrinsic cryoprotective effects. For instance, hyaluronic acid (HA) and trehalose-enriched hydrogels have shown the ability to reduce ice formation and promote post thaw functionality [1,13,14]. Similarly, synthetic polymers such as PEG and PVA have demonstrated ice recrystallization inhibition (IRI) and improved thermal properties [15,16]. Cryogels, microencapsulation techniques, and vitrification protocols are also being adapted to meet the requirements of complex biofabricated tissues [4].

Given the diversity of approaches and the evolving nature of this field, a systematic synthesis of current biomaterials and strategies is essential to inform future development. This review aims to provide a comprehensive overview of the cryoprotective biomaterials used in 3D biofabrication; evaluate their mechanisms of action, structural and functional roles, and compatibility with DMSO-free protocols; compare their advantages and limitations; and highlight emerging trends and future directions in clinical translation.

### Literature Search Strategy

To ensure a comprehensive and representative overview of the field, we performed a structured literature search as described below. We conducted a structured search of peer-reviewed literature published between 2010 and 2025 using PubMed, Scopus, and Web of Science. The search terms included combinations of “cryopreservation”, “biofabrication”, “3D printing”, “biomaterials”, “DMSO-free”, “cryogels”, “microencapsulation”, and specific materials (e.g., “hyaluronic acid”, “PEG”, “gelatin”). Articles were selected based on their relevance to 3D cryopreservation strategies, material characterization, and in vitro or in vivo performance. Priority was given to experimental studies, reviews in high-impact journals, and publications describing clinically relevant or translatable systems.

## 2. Cryoprotective Biomaterials for Three-Dimensional Constructs

### 2.1. Polysaccharide-Based Hydrogels

Polysaccharide-based hydrogels are increasingly prominent in tissue engineering due to their biocompatibility, biodegradability, and structural versatility. These natural polymers, such as HA, chitosan, alginate, and cellulose, mimic the extracellular matrix (ECM), providing a supportive microenvironment for cell adhesion, proliferation, and differentiation. Their high-water content (up to 90%) and tunable mechanical properties achieved through physical or chemical crosslinking enable customization for specific tissue needs, such as load-bearing joints or injectable scaffolds for minimally invasive applications (Table 1).

#### 2.1.1. Hyaluronic Acid

HA is a naturally occurring glycosaminoglycan abundantly found in the ECM, where it regulates essential physiological processes such as cell proliferation, wound healing, tissue hydration, and immune modulation. Its intrinsic biocompatibility, high water retention capacity, and viscoelasticity make it a widely used biomaterial in 3D cell cultures, regenerative medicine, and cryopreservation. Notably, HA can be produced via microbial fermentation, avoiding animal-derived contaminants and supporting clinical-grade applications [17,18].

One of HA’s main strengths lies in its chemical versatility. Derivatization methods such as methacrylation or Pluronic grafting enable the fabrication of hydrogels with tunable mechanical and degradative properties. Moreover, peptide functionalization (e.g., RGD, IKVAV) improves cellular adhesion, migration, and lineage-specific differentiation, supporting applications in glioma modeling, chondrogenesis, and neural repair [19]. Emerging designs, such as cysteamine-grafted HA with reversible disulfide bonds, have yielded self-healing hydrogels with excellent post printing integrity and mechanical resilience, further expanding the biofabrication utility of HA-based platforms [20]. In parallel, HA hydrogels have also proven valuable for tumor spheroid formation and drug testing due to their ability to mimic the tumor microenvironment [17].

In the context of cryopreservation, methacrylated HA (MeHA) hydrogels have shown promising results. For example, MeHA matrices (123 kDa, 54% methacrylation) enabled the homogeneous diffusion of 10% DMSO throughout 3D scaffolds, resulting in post thaw viabilities of 40–60% in human mesenchymal stem cells (MSCs). More importantly, these systems preserved the cells’ differentiation potential, supporting their long-term storage and downstream therapeutic use [37,38]. Additionally, high-molecular-weight HA (HMW-HA, >1 MDa) has been proposed as a non-penetrating macromolecular cryoprotectant that is effective at lowering DMSO requirements. Combinations of 3–5% DMSO with 0.1–0.2% HMW-HA improved the survival and osteo/chondrogenic capacity of MSCs while increasing stemness marker expression (e.g., *CD49f*) [21].

The encapsulation of hMSCs in MeHA matrices has also been linked to the preservation of adipogenic differentiation potential after thawing comparable to that of non-frozen controls, further underscoring the dual structural and bioactive role of HA-based systems in cryopreservation [38]. Furthermore, HA–alginate composite hydrogels (typically in 1:1 ratios) have shown synergistic effects, improving hMSC viability (up to 77.4%), proliferation, and spheroid formation. These systems also maintain key stemness markers (*SOX2*, *OCT4*, *NANOG*) and yield soft porous structures (E ≈ 1.2 kPa), supporting advanced 3D stem cell cultures with regenerative potential [39].

Interestingly, HA also appears to exert intracellular effects during cryopreservation. For instance, its presence in DMSO-containing media attenuated the activation of the RhoA/ROCK pathway, a key mediator of cytoskeletal stress and apoptosis. This suggests that HA’s protective function is not merely physical but also involves the modulation of intracellular signaling [40]. This duality of action, both extracellular and intracellular, positions HA as a unique candidate among natural polymers for next-generation cryopreservation strategies.

#### 2.1.2. Alginate

Alginate is a naturally derived, anionic polysaccharide composed of β-D-mannuronic acid (M) and α-L-guluronic acid (G) residues, primarily extracted from brown algae. Due to its rapid ionic crosslinking with divalent cations such as Ca^2+^, alginate readily forms hydrogels under mild, cell-friendly conditions, making it widely used in tissue engineering, 3D bioprinting, and cryopreservation. The ratio and sequence of M and G blocks determine its mechanical strength, porosity, and gelling behavior, allowing precise control over scaffold architecture and functionality. Although alginate is intrinsically non-adhesive to cells, its bioactivity can be enhanced through functionalization with bioactive peptides such as RGD, YIGSR, or GREDV. These modifications improve cell attachment, proliferation, and lineage-specific differentiation, with demonstrated applications in cardiac and vascular tissue models [19].

In cryopreservation, alginate hydrogels offer several advantages: their hydrated network buffers osmotic fluctuations, limits ice crystal propagation, and promotes the even distribution of CPAs. A notable innovation is the development of core–shell alginate capsules via coaxial electrospraying, which create liquid–core microenvironments that favor 3D cell aggregation and enhanced post thaw recovery. Compared with conventional solid beads, these capsules yielded higher viability and metabolic activity in MSCs when cryopreserved in DMSO–sucrose-based solutions, underlining the relevance of structural design in encapsulation strategies [22].

Further innovations include the use of oxidized and methacrylated alginate (OMA) to fabricate microgels for modular 3D bioprinting. These photopolymerizable hydrogels enable the encapsulation and cryopreservation of human MSCs with a minimal loss of viability and preserved differentiation potential. When printed via the Freeform Reversible Embedding of Suspended Hydrogels (FRESH) technique, OMA microgels yield structurally complex and viable tissue constructs. Their shear-thinning and self-healing behavior facilitates extrusion-based bioprinting while maintaining shape fidelity and scaffold integrity [2].

Encapsulation in alginate has also been used to preserve the structural and functional integrity of complex multicellular systems such as neurospheres. When cryopreserved in ultra-high viscosity alginate crosslinked with Ba^2+^, primary brain neurospheres retained membrane integrity and neuron–astrocyte interactions. The use of CryoStor-CS10 medium and an optimized freezing protocol further improved post thaw recovery, suggesting that alginate encapsulation can maintain both viability and specialized functionality in neural aggregates [11].

However, the intrinsic lack of cell adhesion and the need for chemical modifications to enable functionalization highlight the key limitations of alginate systems. These aspects, together with its slow degradation profile, warrant further research to optimize its long-term biointegration in cryopreserved constructs.

#### 2.1.3. Chitosan

Chitosan is a linear, polycationic polysaccharide derived from the partial deacetylation of chitin. Its biocompatibility, biodegradability, and structural similarity to glycosaminoglycans have made it an attractive biomaterial for various biomedical applications [19].

Chitosan has been applied in cryopreservation as a coating agent in alginate-based encapsulation systems. Its polycationic properties enable the formation of stable polyelectrolyte complexes that enhance capsule integrity and modulate permeability during freezing and thawing. One study demonstrated that probiotic bacteria encapsulated in alginate–chitosan capsules retained higher viability after six months of storage at −20 °C, with preserved functional activity and resistance to gastrointestinal conditions [23]. Another investigation showed that pancreatic islets encapsulated in alginate–chitosan hydrogels maintained post-thaw viabilities exceeding 95%, along with preserved insulin secretion, in contrast to significantly reduced viability in non-encapsulated controls [24]. In line with these findings, the encapsulation of porcine cardiosphere-derived cells (CDCs) in alginate-based matrices preserved cell viability and phenotype over 42 days, while supporting growth factor secretion and enhancing osteogenic differentiation capacity compared with non-encapsulated CDCs, as demonstrated in a recent study [41].

In addition to its use in encapsulation systems, chitosan has also been explored as a structural component in cryogel-based biomaterials. For example, developed sponge-like cryogels composed of chitosan crosslinked with glutaraldehyde were developed using an inverse Leidenfrost effect followed by cryopolymerization at subzero temperatures (−15 °C). Although not directly applied to cellular cryopreservation, these cryogels display key features such as micrometer-sized interconnected pores, rapid swelling, high shape recovery, and mechanical resilience after repeated compression cycles. These properties, along with their biodegradability and tunable porosity, support their potential use as cryocompatible scaffolds facilitating uniform CPA diffusion and mitigating ice damage in 3D construct preservation [25].

#### 2.1.4. Dextran

Dextran is a branched polysaccharide composed of α-1,6-linked D-glucose units, extensively studied for its high hydrophilicity, biocompatibility, and ease of chemical modification. Although lacking inherent cell-binding motifs, dextran can be functionally enhanced through periodate oxidation (Dex-ald), allowing Schiff base crosslinking with amine-containing polymers such as gelatin to form injectable and printable hydrogels under mild conditions [19,26,27].

Dextran-ald has been successfully incorporated into composite bioinks, such as alginate/gelatin/dextran-aldehyde (AGDA), improving structural fidelity and promoting cell viability in both single-cell and spheroid-based constructs [27]. Similarly, photo-crosslinkable systems like hyaluronic acid/dextran-methacrylate (Dex-HEMA) semi-interpenetrating networks offer mechanical robustness and cytocompatibility for long-term culture applications [28].

Dextran-based hydrogels exhibit shear-thinning behavior, rapid crosslinking kinetics, and a high porosity, supporting their use in extrusion-based bioprinting and injectable delivery [29,42]. These properties facilitate homogeneous cell encapsulation and precise filament deposition, while also enhancing cryopreservation performance by promoting uniform cryoprotectant distribution and reducing shear-induced damage. Their hydrophilic network structure can buffer osmotic changes and restrict ice propagation—key factors for maintaining cell viability during freeze–thaw cycles. Additionally, dynamic covalent dextran hydrogels formed via reversible Schiff base chemistry have shown injectability, mechanical robustness, and self-healing behavior under physiological conditions, all of which may support construct stability during cryopreservation [43]. Recent studies have also demonstrated that gelatin–dextran bioinks prepared via non-toxic, catalyst-free crosslinking exhibit fast gelation, shear-thinning behavior, and support homogeneous fibroblast distribution without cytotoxicity [26]. Although their direct application in cryopreservation remains limited, these structural and functional attributes suggest a promising role for dextran-based materials as cryoprotective scaffolds in DMSO-free strategies for complex 3D constructs.

#### 2.1.5. Agarose

Agarose is a thermoresponsive, non-ionic polysaccharide extracted from red algae, composed of alternating D-galactose and 3,6-anhydro-L-galactopyranose residues. It forms a physical gel through reversible hydrogen bonding upon cooling below 35–40 °C, with melting temperatures exceeding 85 °C, making it highly stable for use in biofabrication processes. Due to its bioinert nature and lack of intrinsic cell-adhesive motifs, agarose is often considered a structural scaffold rather than a bioactive matrix [19]. Nonetheless, chemical functionalization, such as the carbodiimide-mediated conjugation of adhesion peptides (e.g., RGD, IKVAV), has enabled the use of agarose hydrogels to support neuronal growth, chondrogenesis, and other lineage-specific cell behaviors [19]. In particular, hybrid hydrogels composed of PEG and agarose have demonstrated enhanced mechanical properties and improved cell encapsulation outcomes, offering a balance between bioinert stability and controlled bioactivity.

Recent findings also support their potential for cryopreservation applications. In a model system using PEG-supplemented agarose hydrogels, encapsulated *E. coli* retained over 80% viability and functionality after three months at −20 °C, supporting the relevance of these hybrid systems for preserving biological constructs under subzero conditions [44]. These systems have also been successfully employed in bone tissue bioprinting, alone or in combination with bioactive ceramics, contributing to in situ mineralization and osteoconductivity [30].

Low melt agarose has also been used as a printable hydrogel for enzyme immobilization in extrusion-based bioprinting. Its fine mesh size allows the high retention of functional proteins while preserving dimensional stability and print fidelity in multilayer constructs [31]. The use of temperature-controlled extrusion systems further enhances reproducibility and allows precise deposition without compromising hydrogel integrity. These characteristics, while explored in enzymatic models, suggest additional utility in cryopreservation settings where functional retention and mechanical stability are critical.

Recent studies have shown that agarose–alginate composite hydrogels (e.g., 3:2 ratio, 5% *w*/*v*) support a high print fidelity without the need for sacrificial supports or photo-crosslinking. These systems demonstrated excellent post printing cell viability (>95%) and sustained chondrogenic function, including glycosaminoglycan production, over 28 days in culture [32]. Their favorable rheological behavior and mechanical resilience make them suitable candidates for integrated printing and preservation workflows. In fact, agarose’s high thermal stability, defined pore structure, and low permeability to macromolecules make it an attractive candidate for use in cryopreservation. It has been employed as a supportive hydrogel matrix for freezing cell-laden constructs, acting as a protective barrier that limits ice crystal formation and reduces mechanical damage during thawing. While its intrinsic cryoprotective effects are limited, its compatibility with cryoprotective agents and structural resilience under thermal cycling enhance its utility in preserving the architecture and integrity of 3D printed scaffolds [19].

#### 2.1.6. Nanocellulose-Based Bioinks

Nanocellulose including cellulose nanofibrils (CNFs) and cellulose nanocrystals (CNCs) has gained considerable attention due to its high mechanical strength, shear-thinning behavior, and chemically versatile surface. CNFs contribute to structural stability in extrusion-based printing, while CNCs enhance scaffold stiffness and crosslinking through hydrogen bonding. In cartilage engineering, nanocellulose–alginate bioinks have demonstrated high shape fidelity (with nozzle diameters < 400 µm) and preserved architecture following cryopreservation [33].

Compared with alginate-only formulations, the inclusion of nanocellulose improves print fidelity, rheological performance, and post thaw structural retention. While Gelatin Methacrylate (GelMA)- and Polyethylene glycol diacrylate (PEGDA)-based bioinks may offer greater bioactivity or tunability, they often require photoinitiators or UV exposure that can compromise cell viability. Nanocellulose-based bioinks, in contrast, provide mechanical reinforcement and cytocompatibility with minimal processing, although they may require functionalization for lineage-specific applications.

Moreover, in photo-crosslinkable systems, nanocellulose improves polymerization kinetics and structural definition. Its incorporation into hybrid bioinks, such as GelMA- methacrylated hyaluronic acid (HAMA) or PEGDA-HA, enhances rheological performance and mechanical strength, supporting long-term cell viability and extracellular matrix preservation [34,35]. This synergy underscores the relevance of nanocellulose as a rheological enhancer and scaffold stabilizer across multiple bioink platforms.

Beyond printability, nanostructured biomaterials also contribute actively to cryopreservation performance. Their finely tuned porosity and stiffness reduce extracellular ice damage and preserve scaffold integrity through freeze–thaw cycles. For example, the combination of CNCs with low-molecular-weight PEG or trehalose has demonstrated enhanced structural preservation and post thaw cell viability, potentially via the stabilization of cell–matrix interactions and the modulation of ice nucleation dynamics [35,36].

While further optimization is needed, particularly in terms of biofunctionalization and cell-instructive signaling, nanocellulose remains a promising additive for cryocompatible bioinks, particularly in applications demanding structural fidelity and mild processing conditions.

### 2.2. Protein-Based Biomaterials

Protein-derived biomaterials such as gelatin, silk fibroin, collagen, and sericin offer exceptional biocompatibility and intrinsic bioactivity due to their resemblance to native ECM components. These proteins contain bioactive motifs (e.g., RGD, GFOGER) that facilitate essential cellular processes like adhesion, proliferation, and differentiation, making them attractive for regenerative medicine, 3D biofabrication, and emerging cryopreservation strategies (Table 2).

#### 2.2.1. Silk Fibroin

Silk fibroin (SF), derived from Bombyx mori, combines exceptional tensile strength with tunable degradation and low immunogenicity. Its structure includes β-sheet crystalline domains that provide mechanical stability and amorphous segments that support cell adhesion [35,45]. SF is shear-thinning and compatible with extrusion and light-based printing methods. Methacrylated SF enables digital light processing (DLP) to fabricate intricate microarchitectures [45]. Incorporation into mesoporous bioglass scaffolds has enhanced compressive strength and osteogenic performance, while blends of mulberry and non-mulberry SF broaden RGD motif diversity, improving hepatic tissue modeling. Structurally, SF features amphiphilic block-like sequences formed by heavy and light chains linked via disulfide bonds. This configuration enables self-assembly and responsiveness to pH and temperature, allowing SF to function as a platform for stimuli-responsive hydrogels, drug carriers, and adhesive materials [46].

In the context of cryopreservation, SF-based hydrogels have demonstrated a protective role in 3D constructs. For example, porous SF cryogels were used to encapsulate epithelial cells in tissue-like models, showing that the scaffold’s mechanical robustness and interconnected porosity preserved post thaw cell–cell and cell–matrix interactions [47]. Notably, SF limited DMSO accumulation within the construct, reducing cytotoxicity while maintaining over 80% cell viability and tight junction integrity after freezing at −80 °C with 5% DMSO. These findings underscore SF’s dual role as both a structural and functional cryocompatible matrix capable of modulating CPA diffusion while preserving cell architecture and viability.

#### 2.2.2. Sericin

Sericin, a globular silk protein co-extracted with fibroin, possesses antioxidant, anti-inflammatory, and epithelial-regenerative properties. Despite its lower mechanical strength compared with fibroin, sericin promotes epithelial proliferation and can be crosslinked with genipin or combined with gelatin to form multifunctional scaffolds with enhanced bioactivity [35,45]. Importantly, sericin exhibits promising cryoprotective properties, including the scavenging of reactive oxygen species (ROS), membrane stabilization, and the maintenance of osmotic balance—mechanisms that parallel those of non-permeating macromolecular cryoprotectants like PEGs. For example, PEG 400 has enabled the successful plunge-freezing of keratinocytes with a post thaw viability comparable to DMSO-based protocols [48], while PEG 200 preserved spheroid integrity via passive diffusion and cytoskeletal stabilization [36]. Recent studies underscore that the inhibition of ice recrystallization during warming, rather than the suppression of initial ice formation, is critical for ensuring cell survival after thawing [49]. Sericin and other structured protein-based hydrogels may thus provide dual functionality, biological support and physical cryoprotection, and hold promise as DMSO-free alternatives in sensitive biomedical applications.

### 2.3. Synthetic Polymers

In contrast to natural polymers, synthetic polymers offer high reproducibility, controlled degradation, and tunable physicochemical properties, making them highly attractive for applications in tissue engineering and cryopreservation. These materials, including PEG, PVA, and poly(lactic-co-glycolic acid) (PLGA), can function as structural scaffolds, drug delivery platforms, or cryoprotective agents, either alone or in combination with natural biomolecules [50,51] (Table 3).

#### 2.3.1. Polyethylene Glycol

PEG is a hydrophilic, non-toxic synthetic polymer widely used in tissue engineering and cryopreservation. Its cryoprotective potential depends on its molecular weight and application mode. Low-molecular-weight PEGs, such as PEG200 and PEG400, have been used in solution as alternative cryoprotectants. PEG400, for instance, enabled the effective plunge-freezing of human keratinocytes, achieving post thaw viabilities comparable to DMSO-based protocols [52]. Similarly, PEG200 allowed the DMSO-free cryopreservation of stem cell spheroids after 2–6 h of preincubation, preserving their viability, proliferative capacity, and cytoskeletal organization [36].

In contrast, high-molecular-weight PEGs and their derivatives (e.g., PEGDA) have been widely explored as hydrogel-forming biomaterials for cryopreservable 3D constructs. PEG hydrogels are non-biodegradable by default but can be chemically modified with degradable segments or acrylate groups, enabling photopolymerization and structural customization. Biofunctionalization with RGD peptides or growth factors improves cell adhesion and ECM production, as demonstrated in mechanically stimulated cartilage constructs [51]. During cryopreservation, PEG-based hydrogels can contribute to osmotic buffering and ice IRI, especially when used in combination with other biomaterials [53].

#### 2.3.2. Polyvinyl Alcohol

PVA is another hydrophilic polymer used in cryopreservation due to its strong ice IRI activity, making it a promising non-permeating additive in cryopreservation protocols. Unlike permeating agents such as DMSO, PVA does not require intracellular diffusion to exert its protective effect. Instead, it adsorbs to ice surfaces and suppresses crystal growth during thawing, thereby reducing mechanical damage and improving post thaw cell viability and morphology [15].

Although most studies have evaluated PVA as a solute in cryoprotective solutions, it also has potential as a structural component in hydrogel systems. Physically crosslinked PVA hydrogels, produced by freeze–thaw cycles, have been explored in tissue engineering for their mechanical robustness, water retention, and cytocompatibility [50]. These properties suggest that PVA-based hydrogels could serve as dual-function platforms offering both mechanical support and cryoprotection for 3D biofabricated constructs.

#### 2.3.3. Polyurethane (PU)

PU is a versatile class of synthetic polymers composed of soft and hard segments linked by urethane bonds. Recent developments in biodegradable, thermosensitive PU hydrogels have enabled their use in 3D printing for neural and vascular tissue engineering. These materials exhibit a favorable mechanical strength, elasticity, and tunable degradation. Low-temperature deposition techniques have been employed to fabricate PU-based constructs incorporating adipose-derived stem cells and cryoprotectants, with constructs maintaining high post thaw cell viability and structural integrity after freeze–thaw cycles [51].

Collectively, synthetic polymers are emerging not only as passive carriers or structural supports but also as active agents in modulating the physicochemical environment during cryopreservation. Thanks to their modular structure, these polymers are well suited for the design of next-generation bioinks and cryoprotective systems tailored for regenerative medicine [15,50,51,53].

### 2.4. Comparative Insights Across Cryoprotective Biomaterials

Given the diversity of biomaterials investigated for the cryopreservation of 3D constructs, a comparative analysis is essential to understand their respective advantages, limitations, and potential synergies. Natural polymers, such as polysaccharides and proteins, provide excellent biocompatibility and ECM mimicry, but often require structural reinforcement or functionalization. Synthetic polymers offer tunable physicochemical properties, mechanical robustness, and batch-to-batch reproducibility, yet lack inherent bioactivity unless modified. Table 4 summarizes the key differences across representative biomaterials used in DMSO-free or low-toxicity cryopreservation strategies.

Polysaccharide-based hydrogels like HA and alginate support osmotic buffering, a high water content, and structural integrity during freezing, but may be mechanically weak or lack adhesive motifs unless combined with other polymers or peptides [17,18,19,20,21,22]. Chitosan and dextran provide structural stabilization and capsule reinforcement, while nanocellulose excels in shape fidelity and rheological control, albeit with limited intrinsic bioactivity [23,24,25,33,35].

On the other hand, protein-based materials such as silk fibroin and gelatin offer superior bioactivity and mechanical performance, with fibroin enabling DMSO reduction and structural preservation in 3D constructs [35,45,47]. Sericin, while mechanically weaker, adds antioxidant and membrane-stabilizing properties, positioning it as a biologically active additive for DMSO-free applications [35,45,49]. Synthetic polymers like PEG, PVA, and PU provide exceptional tunability and processability. PEG-based hydrogels enable osmotic buffering and CPA delivery, while low-molecular-weight PEGs and PVA exert ice IRI without penetrating cells [36,50,52]. PU-based cryogels show promising viability and structural fidelity after freeze–thaw cycles, though long-term biocompatibility remains under investigation [51].

While no single biomaterial satisfies all ideal criteria for 3D cryopreservation, hybrid systems that combine ECM-mimetic properties with mechanical stability and IRI activity are emerging as promising candidates. Composites such as HA–alginate, PEG–agarose, or gelatin–dextran exemplify how synergistic designs can overcome the limitations of individual materials and support the clinical translation of cryopreservable tissue constructs [19,26,27,39].

## 3. Cryopreservation in 3D Constructs: From Non-Bioprinted to Bioprinted Systems

### 3.1. Microencapsulation and Microspheres

Microencapsulation has emerged as a promising strategy to enhance the cryopreservation of therapeutic cells by replicating the features of the native extracellular matrix. Hydrogel-based microcapsules provide a semi-permeable barrier that buffers cells from osmotic stress and mechanical damage while allowing controlled nutrient and gas exchange post thaw. Compared with conventional suspension systems, encapsulated constructs offer an improved retention of viability and function; however, challenges such as non-uniform CPA diffusion, thermal heterogeneity, and matrix deformation during freezing and thawing persist (Figure 1).

Among the various encapsulation materials, core–shell architectures composed of alginate hydrogels have demonstrated a notable cryoprotective performance. Thus, mesenchymal stem cells encapsulated in such microspheres exhibited post thaw viabilities above 70% following vitrification with low CPA concentrations. In addition to promoting survival, the encapsulation strategy preserved stemness and multilineage differentiation potential, supporting its scalability for regenerative medicine applications [36]. Hybrid systems incorporating chitosan with alginate further improve structural integrity and reduce CPA permeability. The observed enhancements in post thaw viability and metabolic activity are attributed to more effective ice control and the improved stabilization of the cell–matrix interface [24].

Importantly, pre-conditioning strategies can substantially influence cryopreservation outcomes. For example, MSCs subjected to chondrogenic pre-differentiation prior to freezing retained the capacity to synthesize glycosaminoglycans and collagen post thaw—an effect of particular relevance for applications in cartilage or intervertebral disk repair [54]. These results emphasize that the physiological status of the encapsulated cells prior to cryopreservation is a critical variable that deserves greater attention in protocol design.

Another promising avenue involves the use of ice-nucleating agents (INAs) to reduce the risks associated with intracellular ice formation. By triggering ice crystallization at higher subzero temperatures, INAs such as crystalline cholesterol mitigate supercooling and the associated mechanical damage. In alginate-encapsulated liver spheroids, INA-based cryopreservation significantly enhanced post thaw viability, metabolic activity, and key hepatic functions, including albumin and cytochrome P450 expression [55]. This suggests that INA-assisted cryopreservation protocols may be particularly suitable for “ready-to-use” microtissue therapies.

The growing interest in the cryopreservation of complex microstructures such as encapsulated spheroids and microtissues has introduced new challenges, including heterogeneous cell density, matrix variability, and limited diffusional uniformity. To overcome these limitations, recent approaches emphasize the rational design of the encapsulating matrix, including the modulation of porosity and mechanical stiffness to improve cryoprotectant penetration and ice management during freezing. These hydrogel design strategies are often complemented by the incorporation of disaccharides like trehalose and the use of slow-permeating cryoprotectants, aiming to achieve a homogeneous CPA distribution and controlled thawing kinetics. Together, these parameters are increasingly recognized as critical for maintaining post thaw viability and functional performance in complex 3D constructs [56].

Beyond encapsulation, scaffold-free strategies based on spheroid assembly also face specific cryopreservation challenges. Ensuring consistent spheroid morphology and size is critical, as variability in diameter or roundness may lead to mechanical misalignment during fabrication, ultimately compromising tissue geometry, maturation, and function [57]. Furthermore, heterogeneous CPA distribution and thermal gradients during freezing can promote uneven ice crystal formation and localized damage, reducing post thaw viability and structural fidelity. To overcome the limitations of scaffold-free cryopreservation, future strategies must not only optimize spheroid size, shape, and CPA delivery, but also address the critical phase of rewarming. Devitrification and ice recrystallization during rewarming can severely compromise structural and cellular integrity, particularly in volumetric constructs. To mitigate these effects, advanced rewarming technologies such as photothermal heating (e.g., gold nanorods, graphene oxide) and magnetic induction heating using iron oxide nanoparticles have been developed. These approaches provide rapid and homogeneous rewarming, reducing thermal gradients and preserving both scaffold architecture and cell function post thaw [58]. Their integration into cryopreservation workflows represents a promising avenue for improving viability and functionality in both encapsulated and scaffold-free microtissues (Figure 1).

### 3.2. Macroscopic Hydrogels/Scaffolds Not Fabricated by Bioprinting

Macroscopic hydrogels fabricated through non-bioprinting strategies such as molding, casting, freeze-drying, and robotic dispensing constitute a versatile platform for tissue engineering and regenerative medicine. These constructs enable the formation of 3D microenvironments that closely mimic native ECM, while supporting nutrient transport, mechanical integrity, and cell–matrix interactions [59,60].

Hydrogels can be classified by their crosslinking mechanism as either physically or chemically crosslinked, and by their origin as natural (e.g., alginate, collagen, chitosan, HA) or synthetic (e.g., PEG, PVA, PU). Each class offers distinct advantages: natural polymers provide inherent bioactivity and degradability, while synthetic ones offer mechanical tunability, reproducibility, and chemical stability. Composite and hybrid hydrogels are designed to integrate the bioactivity of natural polymers with the mechanical tunability of synthetic materials, thereby enhancing cryostability, structural stiffness, and controlled degradation [61,62].

Porosity and interconnectivity are critical design parameters in these scaffolds, as they influence nutrient diffusion, cell infiltration, and CPA transport. Fabrication techniques such as freeze-drying, particulate leaching, and gas foaming allow the generation of porous scaffolds with a tunable architecture. However, limited thermal conductivity and poor CPA penetration in larger constructs can impair cryopreservation, leading to ice formation, osmotic damage, and viability loss particularly in constructs exceeding 200 µm in thickness [60,62,63].

To address these challenges, modular approaches have incorporated microencapsulated cells into macroscopic scaffolds. For instance, the robotic dispensing of cell-laden microbeads within alginate struts enabled the construction of large constructs (20 × 20 × 6 mm) with interconnected channels and preserved cell viability (>90%) after 7 days in culture [64]. This strategy reduced shear-induced cell damage and facilitated localized protection during freezing.

Further improvements include the use of macroporous cryogels, which combine elasticity, shape memory behavior, and convective permeability. Their open pore networks facilitate CPA exchange and minimize intracellular ice formation during freezing–thawing cycles, enhancing cell survival. These materials have been explored as injectable scaffolds compatible with cryopreservation workflow [59]. Decellularized extracellular matrix (dECM)-derived scaffolds are also emerging as cryopreservable macroscopic constructs. While they retain tissue-specific biochemical cues, they pose unique challenges due to their heterogeneous composition and limited control over porosity. Their preservation demands tailored protocols to balance bioactivity retention and structural integrity [63]. In summation, macroscopic hydrogels not fabricated by bioprinting remain essential platforms for regenerative applications. Their successful cryopreservation hinges on the optimization of pore structure, mechanical resilience, and CPA diffusion, which collectively determine post thaw viability and scaffold functionality.

### 3.3. Bioprinted Constructs: Challenges and Emerging Strategies

The cryopreservation of bioprinted tissues poses distinctive technical and biological challenges, largely due to their spatial complexity and functional requirements. Unlike conventional 2D cultures or non-printed 3D constructs, bioprinted tissues demand the preservation of not only cellular viability but also architectural fidelity, interlayer cohesion, and anisotropic mechanical properties. Major obstacles include intracellular ice formation, osmotic imbalance, and cryoprotectant toxicity issues that become more pronounced in thick, cell-dense constructs.

The evolution of bioprinting technologies has significantly influenced cryopreservation strategies, particularly vitrification. The ability to fabricate reproducible architectures with tunable porosity and defined geometry enables more predictable thermal profiles, controlled CPA diffusion, and reduced construct thickness, key parameters for ice-free cryopreservation. Advances in polysaccharide-based bioinks have further enhanced mechanical stability and cytocompatibility, facilitating the design of constructs better suited to withstand freezing and thawing stresses [65]. Moreover, the development of multilayered tissue constructs and precise layer-by-layer cell deposition [66] has not only advanced tissue engineering but also provided reliable in vitro models to optimize cryopreservation protocols in complex 3D systems.

To address these challenges, various technological and material-based strategies have been proposed. Among them, temperature-controlled cryoprinting (TCC) represents an emerging technique that integrates layer-by-layer bioprinting with simultaneous cryogenic solidification. In this method, the printed construct descends progressively into a cooling medium, ensuring controlled thermal gradients and minimizing ice crystal formation [67]. Although promising, TCC remains under early-stage investigation and has been primarily explored by individual research groups.

Alternative cryogenic printing platforms, such as those employing dry ice and isopropanol baths, have shown the ability to preserve soft, complex geometries with high cellular viability [68]. These strategies highlight the importance of thermal control, hydrogel composition, and crosslinking to preserve both structure and function during freezing.

In parallel, the intrinsic properties of bioinks are increasingly leveraged to support cryopreservation. Hydrogels composed of alginate, agar, HA, or PVA can provide thermal insulation, mechanical resilience, and cytocompatibility. For example, alginate–agar blends used in cryoprinting systems exhibit favorable viscoelastic responses during freezing, minimizing structural deformation and supporting cell function [69].

As the field progresses, integrating these material-based strategies with precise thermal control and biomanufacturing scalability will be essential to advance cryopreservation-ready bioprinted constructs toward clinical translation.

### 3.4. Overcoming Cryopreservation Challenges in 3D Systems: Safe Cryoprotectants and Effective Protocols

DMSO remains the most commonly used cryoprotectant due to its high membrane permeability and ability to prevent intracellular ice formation. However, its application is limited in clinical and biofabrication contexts because of its cytotoxicity, pro-inflammatory potential, and challenges in complete removal from 3D scaffolds. DMSO induces osmotic stress, leading to membrane destabilization and reduced post thaw viability—particularly critical in 3D constructs where the cryoprotectant distribution is heterogeneous [70]. Furthermore, it has been associated with increased oxidative stress, mitochondrial dysfunction, and apoptotic signaling pathways [70]. These effects are exacerbated in bioinks, where DMSO interferes with crosslinking agents like calcium chloride, leading to decreased viscosity and compromised structural integrity [9].

These limitations have catalyzed intensive research into alternative cryoprotectants with improved biocompatibility and functionality in 3D systems. Among them, trehalose, alone or in combination with ethylene glycol or glycerol, has emerged as a widely studied sugar-based cryoprotectant. It preserves the viability, genomic stability, and pluripotency of stem cells while minimizing genotoxicity and ER stress, features that are desirable for GMP-compliant applications [71]. To overcome its limited membrane permeability, advanced delivery methods such as cold-responsive nanocapsules, membrane-permeating peptides, and electroporation have been employed to enhance intracellular accumulation and cryoprotective efficacy [58]. In parallel, several biomaterials have shown potential as DMSO-free alternatives. Hydrogels based on hyaluronic acid–sericin blends, glycopolymers, PVA, and polyampholytes have demonstrated high post thaw viability and cytocompatibility in stem cell-based constructs. These materials modulate ice nucleation, reduce osmotic stress, and can be integrated into emerging cryobioprinting workflows.

Beyond cryoprotectant selection, scaffold architecture plays a decisive role in cryopreservation outcomes. Geometric parameters such as pore size, lattice interconnectivity, and scaffold thickness influence not only nutrient diffusion and mechanical performance but also CPA penetration and thermal conductivity during freezing and thawing. For instance, layered alginate scaffolds co-printed with mesenchymal stem cells supported hepatocyte aggregation and maintained functional gene expression post printing [72].

Regarding the cryopreservation protocols in 3D systems, ranging from microscale encapsulated spheroids to macroscale cryogels, two main approaches are employed: slow freezing and vitrification. Unlike in 2D cultures, applying these strategies to 3D constructs introduces unique challenges, including CPA diffusion gradients, spatially heterogeneous cooling, and construct-dependent thermal dynamics. Protocol optimization must therefore be tailored to cell type, scaffold composition, construct geometry, and surface-to-volume ratio.

Slow freezing involves gradual cooling (1 °C/min) in the presence of permeating CPAs such as DMSO or glycerol, promoting cellular dehydration and minimizing intracellular ice formation. However, this method is prone to solution effect injury, caused by rising solute concentrations and osmotic imbalance during ice formation. Controlled-rate cooling, ice nucleation (seeding), and optimized CPA loading protocols can mitigate these effects, especially in cell clusters and tissue fragments [73]. The incorporation of zwitterionic CPAs in slow freezing protocols has also demonstrated enhanced post thaw viability and functional preservation in complex tissue constructs, including spheroids and patient-derived tumor xenografts [74].

Vitrification, in contrast, prevents ice formation by inducing a glass transition through ultra-rapid cooling and high CPA concentrations. Water molecules lose mobility before ice nucleation can occur, forming an amorphous solid that preserves both the cellular and extracellular architecture [75]. This technique prevents mechanical damage from crystallization and limits osmotic imbalance, making it especially valuable for preserving delicate or highly structured tissues. Nevertheless, the elevated CPA concentrations required for vitrification increase the risk of chemical toxicity and osmotic stress, necessitating precise protocol design and rapid execution. Vitrification plays a central role in modern cryopreservation, especially in fields like reproductive medicine, where it has consistently outperformed slow freezing. Meta-analyses show higher post thaw survival rates and clinical outcomes for vitrified oocytes and embryos [76,77]. In the context of 3D tissue constructs, vitrification becomes more complex due to diffusion limitations, thermal heterogeneity, and construct thickness, which challenge uniform CPA distribution and cooling efficiency. Recent efforts include the use of miniaturized constructs, ice-blocking agents, and nano-warming techniques to facilitate uniform vitrification while minimizing toxicity.

In conclusion, both cryopreservation strategies offer valuable tools for preserving 3D cellular constructs, yet each carries intrinsic limitations (Figure 2). Slow freezing is widely applicable and scalable but is constrained by ice-related damage and diffusion gradients in larger constructs. Vitrification provides superior structural preservation, particularly in microencapsulated or miniaturized systems, but demands strict protocol control and carries a higher cytotoxicity burden. Hybrid approaches, including optimized CPA formulations, progressive loading strategies, and computationally guided thermal protocols, are under investigation to reconcile these challenges and support the safe, scalable cryopreservation of engineered tissues.

## 4. Future Perspectives and Regulatory Challenges

The successful cryopreservation of 3D biofabricated constructs is increasingly recognized as a cornerstone for the clinical deployment of regenerative medicine therapies. The convergence of cryobiology, biomaterials science, and 3D printing technologies is enabling the development of integrated platforms for the long-term storage, transport, and on-demand application of engineered tissues. However, significant challenges remain, and the transition from experimental success to clinical reality will depend on future innovations, regulatory alignment, and scalable methodologies.

One key area of advancement lies in the reduction in cryoprotectant-associated cytotoxicity. Conventional agents such as DMSO, though widely used, present toxicity and regulatory limitations that hinder their use in therapeutic products [70]. Emerging DMSO-free alternatives, including trehalose-based systems, polyampholytes, and synthetic polymers such as polyglycerol methacrylate (PMO), offer improved biocompatibility, intracellular protection, and ice IRI [49,71,78]. While many of these agents remain underexplored in 3D systems, their mechanisms of action provide a solid foundation for adaptation to complex constructs.

In parallel, biomaterials research continues to expand the portfolio of cryocompatible scaffolds. Smart and stimuli-responsive hydrogels, such as methacrylated bioinks and thermosensitive polymers like Pluronic F127, enable spatially controlled crosslinking and improved cryoprotectant distribution [79,80]. Composite hydrogels combining natural polysaccharides (e.g., hyaluronic acid, alginate, dextran) with synthetic components (e.g., PEG, PVA) or nanomaterials (e.g., nanocellulose, graphene oxide, laponite) have demonstrated superior print fidelity, mechanical stability, and post thaw structural integrity [81,82].

These advances are particularly relevant for clinical applications involving skin grafts, cartilage repair, and organoid-based disease models, where post thaw structural integrity and cellular functionality are critical. For instance, several hydrogel systems, including GelMA-based scaffolds, chitosan–hydroxyapatite composites, and vitrified tumor spheroids, are being investigated for use in musculoskeletal regeneration, craniofacial repair, and personalized oncology models, respectively.

Among these materials, chitosan-based scaffolds hold particular promise due to their chemical versatility, biocompatibility, and ability to form hydrogels through pH- or temperature-responsive gelation. Recent developments include photoChito, a methacrylated chitosan derivative designed for light-based crosslinking, which allows high-resolution bioprinting and rapid gelation under mild conditions. When combined with recombinant collagen, photoChito-based bioinks have demonstrated excellent printability and >90% epithelial cell viability in 3D skin constructs [83]. While its application in cryopreservation has not yet been validated, its photo-crosslinkable nature and structural fidelity under physiological conditions suggest potential compatibility with layered or cryo-adapted bioprinting workflows, especially when combined with less cytotoxic cryoprotectants. In parallel, thermosensitive chitosan–CNC composites incorporating β-glycerophosphate allow for in situ gelation at physiological temperatures, offering injectability, mechanical support, and cryo-relevant adaptability [84,85]. Moreover, multifunctional chitosan-based scaffolds enriched with riboflavin or hydroxyapatite exhibit antioxidant and antibacterial activity that could mitigate oxidative stress and inflammation post thaw [86]. Although further cryopreservation-specific validation is required, these formulations represent promising candidates for integration into advanced DMSO-free or hybrid cryopreservation strategies. Despite this potential, they remain largely untested under standardized freezing–thawing conditions. Bridging this gap should be a key focus for future research, particularly in evaluating their behavior in DMSO-free protocols and clinically relevant cryopreservation cycles.

Similarly, gelatin-based hydrogels such as GelMA, when blended with alginate, PEGDA, or nanoclays, offer tunable degradation profiles and enhanced mechanical strength, making them well-suited for cryopreservable architectures targeting musculoskeletal or soft tissue regeneration [35,87,88,89]. Collagen and collagen-like peptides (CLPs), although underutilized in cryopreservation contexts, offer additional opportunities due to their ECM-mimetic structure, self-assembling properties, and capacity for hybridization with damaged matrices. These scaffolds could support both structural stability and bioactive signaling during freeze–thaw cycles [46,90]. Synthetic scaffolds such as PLGA and polycaprolactone (PCL) are also being explored for their mechanical robustness and adaptability to cryogenic workflows. PLGA, used successfully in wound healing models and functionalized with RGD peptides or hyaluronic acid, has shown enhanced cell interaction and regenerative capacity [91]. Although PCL has not yet been applied in cryopreservation settings, its excellent printability and mechanical strength suggest its suitability as a structural backbone for hybrid cryocompatible scaffolds [51].

On the other hand, technological advances are further driving the integration between biofabrication and preservation. TCC, cryopreservation-adapted bioprinters, and decentralized biobanking systems are converging to form unified, automated platforms for fabrication and storage. These systems enable GMP-compatible workflows with real-time quality control, which is essential for transitioning to clinical-grade tissue production [67]. However, persistent barriers still remain. Standardized vitrification protocols optimized for 3D constructs are scarce, and the limited penetration of cryoprotectants into large or dense scaffolds increases the risk of cryoinjury. Furthermore, current strategies rarely account for the architectural complexity and layer-specific heterogeneity characteristics of bioprinted tissues [67,75]. Addressing these limitations will require a multidisciplinary approach integrating material sciences, cryobiology, engineering, and regulatory science.

From a translational perspective, future efforts must prioritize the development of reproducible, scalable, and regulatory-compliant cryopreservation workflows. This includes the use of GMP-grade biomaterials, the validation of storage and transport conditions, and the establishment of standardized quality control metrics particularly for assessing post thaw cell viability, mechanical functionality, sterility, and biocompatibility. Regulatory frameworks must evolve to accommodate the specific challenges of cryopreserved 3D constructs, such as batch variability, long-term stability, and combination product classification. In summary, the cryopreservation of 3D biofabricated constructs is poised to become a foundational component of regenerative medicine. The integration of advanced biomaterials, non-toxic cryoprotectants, and smart fabrication platforms is transforming the landscape, bringing clinically viable engineered tissues closer to reality. Achieving this goal will require sustained interdisciplinary collaboration and regulatory innovation to deliver robust, off-the-shelf therapies that meet the complex demands of precision medicine.

## Figures and Tables

**Figure 1 ijms-26-06908-f001:**
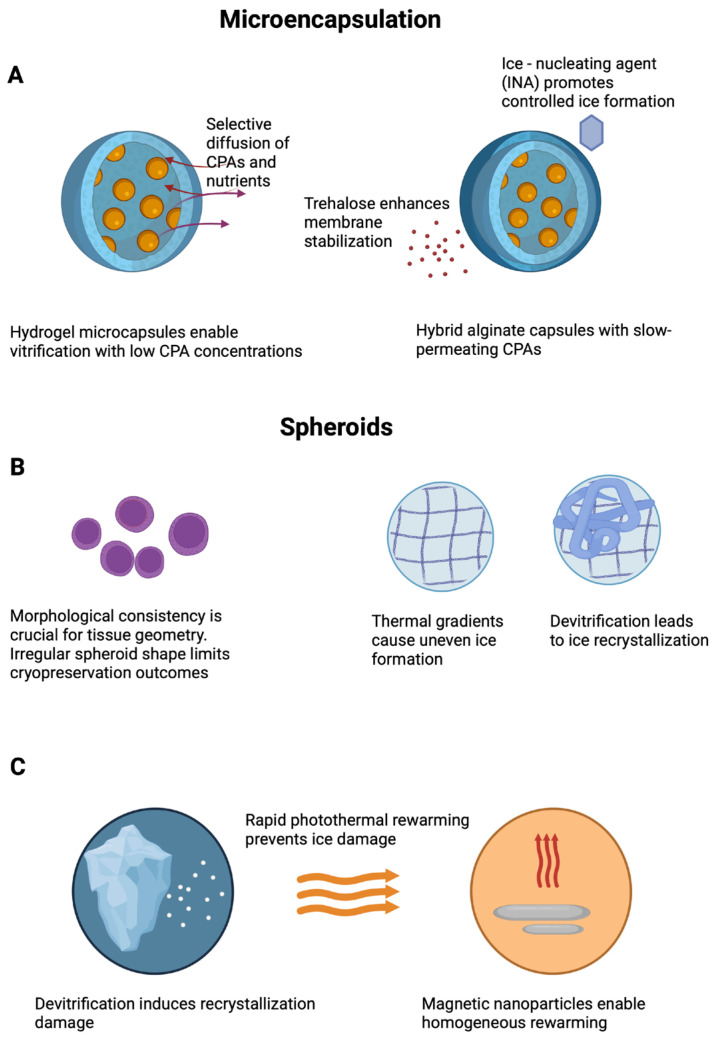
Cryopreservation strategies and failure mechanisms in microencapsulated and scaffold-free 3D constructs. (**A**) Hydrogel microcapsules improve vitrification by modulating CPA diffusion and membrane stability. (**B**) Spheroids exhibit variable cryostability due to morphological irregularities and thermal gradients. (**C**) Advanced rewarming methods, including photothermal and magnetic nanoparticle-based strategies, minimize recrystallization risks. (Created with BioRender).

**Figure 2 ijms-26-06908-f002:**
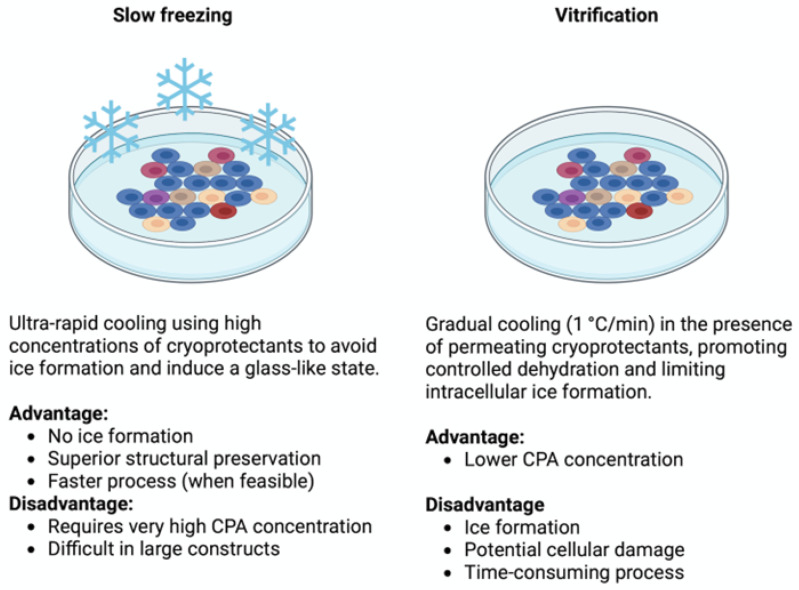
Comparison between slow freezing and vitrification strategies for cryopreservation of 3D constructs (created with BioRender).

**Table 1 ijms-26-06908-t001:** Polysaccharide-based cryoprotective biomaterials. Representative natural hydrogels with cryoprotective functions, key physicochemical properties, and 3D biomedical applications.

Material Type	Examples	Key Properties	Cryoprotective Function	Applications	References
**Hyaluronic acid (HA)**	MeHA, HMW-HA	Viscoelastic, ECM-mimetic, chemically modifiable	Uniform CPA diffusion, maintains differentiation	MSCs, tumor models	[17,18,19,20,21]
**Alginate**	Ca^2+^-crosslinked, OMA	Fast gelation, functionalizable	Encapsulation, electrosprayed core–shell systems	Neural spheroids, biofabricated constructs	[2,11,22]
**Chitosan**	Alginate–chitosan capsules, cryogels	Polycationic, biodegradable, stabilizing	Capsule reinforcement, modulates permeability, cryogel matrix	Cell encapsulation, islet preservation, cryoscaffolds	[23,24,25]
**Dextran**	Dex-ald, Dex-HEMA	Injectable, photopolymerizable	Potential osmotic buffering, scaffold structure	Long-term cultures, DMSO-free systems	[26,27,28,29]
**Agarose**	Low melt, composites	Thermally stable, bioinert	Ice crystal barrier, structural integrity	Cartilage engineering, enzyme scaffolds	[19,30,31,32]
**Nanocellulose**	CNFs, CNCs	High stiffness, shear-thinning	Prevents freeze–thaw damage, ECM stability	Hybrid bioinks, cryoprinting	[33,34,35,36]

**Table 2 ijms-26-06908-t002:** Protein-based cryoprotective biomaterials. Summary of protein-derived materials with cryoprotective roles, physicochemical features, and applications in tissue engineering and DMSO-free systems.

Material Type	Examples	Key Properties	Cryoprotective Function	Applications	References
**Silk Fibroin** **(SF)**	Methacrylated SF	Shear-thinning, β-sheet stability, pH/temp-responsive	Supports cell–cell/matrix interaction, reduces DMSO toxicity	Tissue engineering, hepatic modeling, cryogel interactions	[35,45,46,47]
**Sericin**	Genipin-crosslinked	Antioxidant, epithelial regenerative, bioadhesive	ROS scavenging, membrane stabilization, osmotic buffering	Epithelial scaffolds, cryopreservation without DMSO	[36,48,49]

**Table 3 ijms-26-06908-t003:** Synthetic cryoprotective polymers. Key properties, cryoprotective roles, and biomedical uses of representative synthetic polymer-based materials.

Material Type	Examples	Key Properties	Cryoprotective Function	Applications	References
**Polyethylene glycol** **(PEG)**	PEG200, PEG400, PEGDA	Hydrophilic, modifiable, non-toxic, crosslinkable	Intracellular protection, osmotic buffering, ice inhibition	Cryopreservatio, hydrogel bioinks	[36,51,52,53]
**Polyvinyl alcohol (PVA)**	Freeze–thaw hydrogels	Physically crosslinked, IRI-active, non-permeating	Suppresses ice growth, enhances morphology and post thaw viability	Cryopreservation additives, tissue scaffolds	[15,50]
**Polyurethane (PU)**	Thermoresponsive PU	Elastic, biodegradable, tunable stiffness	Maintains structure/function during freeze–thaw cycles	Neural/vascular tissue printing, injectable cryogels	[15,50,51,53]

**Table 4 ijms-26-06908-t004:** Comparative properties of cryoprotective biomaterials. Overview of natural and synthetic materials for 3D cryopreservation, highlighting key functions, printability, structural features, and limitations.

Biomaterial	Cryoprotective Function	Printability	Stability	Remarks/Limitations	References
**Hyaluronic acid (HA)**	ECM mimicry, osmotic buffering	Tunable (e.g., MeHA)	Low	Often combined with stronger polymers	[17,18,19,20,21]
**Alginate**	Osmotic buffering, water retention	Excellent (ionic crosslinking)	Moderate	Requires peptide modification	[2,11,22]
**Chitosan**	Membrane stabilization, capsule reinforcement	Limited (requires acidic pH)	Moderate	Complex gelation, used in encapsulation	[24,25,41]
**Dextran**	IRI potential, CPA diffusion	Shear-thinning, injectable	Moderate	Suitable for injectable cryogels	[26,27,28,29]
**Agarose**	Thermal stability, structural retention	Limited (thermo-gelling)	High	Supports architecture retention; lacks cell interaction	[19,30,31,32]
**Nanocellulose** **(CNC, CNF)**	Rheological tuning, shape fidelity	High (in blends)	High	Bioinert unless functionalized	[33,36]
**Silk fibroin**	DMSO reduction, membrane protection	Tunable (physical/chemical)	High	Good mechanical properties post thaw	[35,45,46,47]
**Sericin**	Antioxidant, membrane-protective	Blendable, injectable	Low	Bioactive additive with limited mechanical support	[36,48,49]
**PEG/PEGDA**	CPA delivery, osmotic buffering	High (photo/chemical)	Good	Synthetic, tunable stiffness and gelation	[36,51,52,53]
**PVA**	Ice recrystallization inhibition (IRI)	Blendable	High	Inert, used as non-toxic additive	[15,50]
**PU**	Elastic cryogelation, shape memory	Cryogel-based	High	Promising for scaffold integrity post thaw	[15,50,51,53]

## Data Availability

No new data were created or analyzed in this study. Data sharing is not applicable to this article.

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
