# Peer review of "Advances in Cryopreservation Strategies for 3D Biofabricated Constructs: From Hydrogels to Bioprinted Tissues"

_ijms, 2025, doi:10.3390/ijms26146908_

Round 1
Reviewer 1 Report
Comments and Suggestions for Authors
In this manuscript, the authors reviewed the current strategies for the cryopreservation of 3D constructs and examine innovative approaches such as temperature-controlled cryoprinting, nanowarming, and hybrid scaffolds with enhanced cryocompatibility. They also discussed the regulatory and manufacturing challenges associated with developing GMP-compliant cryopreservation workflows. Overall, the manuscript provides an integrated perspective on material-based strategies for 3D cryopreservation and identifies future directions to enable the long-term storage and clinical translation of engineered tissues. However, some problems should be revised before considering publication, as follows:
1、 Please add some specific clinical application cases in the introduction.
2、 Lines 45–47, 53, and 372 contain unclear phrasing. A thorough revision is recommended to improve clarity and ensure precise expression across the manuscript.
3、 While the authors report high cell viability, a comparison with non-encapsulated controls or other encapsulation methods is lacking. Line 181-185.
4、 Lines 190-193, please add some specific effects or advantages of this material in cryopreservation.
5、 Lines 211–212, more supporting references, especially those detailing underlying mechanisms and experimental evidence, are needed to validate these claims related to cryopreservation.
6、 It is better to emphasize the innovation of PEG and agarose composite hydrogels in cryopreservation. Line 229-230.
7、 It is recommended to discuss the advantages and limitations of the described bioink in comparison with other types of bioinks to provide a clearer context for its potential applications.
8、 A comparative discussion on the advantages and limitations of this bioink relative to other types is recommended to enhance the clarity and significance of the findings. Line 259-261.
9、 Figure 1 needs punctuation and clear separation to distinguish different concepts in the figure description.
Author Response
Dear reviewer,
We have addressed your comments including point-by-point the response to the concerns that you can find next. All the changes made in the manuscript are highlighted in yellow:
“In this manuscript, the authors reviewed the current strategies for the cryopreservation of 3D constructs and examine innovative approaches such as temperature-controlled cryoprinting, nanowarming, and hybrid scaffolds with enhanced cryocompatibility. They also discussed the regulatory and manufacturing challenges associated with developing GMP-compliant cryopreservation workflows. Overall, the manuscript provides an integrated perspective on material-based strategies for 3D cryopreservation and identifies future directions to enable the long-term storage and clinical translation of engineered tissues. However, some problems should be revised before considering publication, as follows:”
1.Please add some specific clinical application cases in the introduction.
We appreciate this helpful suggestion. We have added specific clinical examples to the introduction, including the cryopreservation of 3D-printed cartilage grafts and hepatic microtissues used in regenerative medicine and transplant models. These cases provide real-world context for the relevance of cryopreservation strategies. Please see lines 48-53 in the revised manuscript.
Included text:
“Applications include off-the-shelf tissue implants, patient-specific models for personalized medicine, and advanced therapeutic products such as bioprinted corneal epithelium and osteochondral grafts [3] [4] [5]. Other clinically oriented examples include cryopreserved hepatic microtissues used in drug toxicity testing and liver disease modeling, further emphasizing the need for cryopreservation protocols that ensure post-thaw structural and functional integrity [6].”
- Lines 45–47, 53, and 372 contain unclear phrasing. A thorough revision is recommended to improve clarity and ensure precise expression across the manuscript.
Line 45-47 and 53:
Response:
Thank you for this valuable comment. As part of the major revision, the original sentences at lines 45 - 47 and 53 have been restructured and replaced as part of an expanded and reworked introduction, following both your feedback and additional guidance from Reviewer 2. These changes aimed to enhance clarity, define the scope more precisely, and incorporate clinically relevant examples.
Line 372:
Original: Their molecular modularity makes them ideal candidates for the next generation of bioinks and cryoprotective systems in regenerative medicine.
Revised: line 406-408.
Thanks to their modular structure, these polymers are well-suited for the design of next-generation bioinks and cryoprotective systems tailored for regenerative medicine [15] [53] [54] [56].
- While the authors report high cell viability, a comparison with non-encapsulated controls or other encapsulation methods is lacking. Line 181-185.
We thank the reviewer for this valuable observation. We have revised the text to highlight a more explicit comparison between encapsulated and non-encapsulated systems. In addition to the previously cited study on pancreatic islets, we now include findings from our own published work, where encapsulated porcine cardiosphere-derived cells (CDCs) demonstrated enhanced viability, sustained phenotype, and superior osteogenic differentiation compared to non-encapsulated counterparts. This update reinforces the advantages of encapsulation for cryopreservation and functional performance.
Original: Another investigation showed that pancreatic islets encapsulated in alginate–chitosan hydrogels maintained post-thaw viabilities exceeding 95%, along with preserved insulin secretion, in contrast to significantly reduced viability in non-encapsulated controls [18].
Revised (line 190-197): Another investigation showed that pancreatic islets encapsulated in alginate–chitosan hydrogels maintained post-thaw viabilities exceeding 95%, along with preserved insulin secretion, in contrast to significantly reduced viability in non-encapsulated controls [25]. In line with these findings, encapsulation of porcine cardiosphere-derived cells (CDCs) in alginate-based matrices preserved cell viability and phenotype over 42 days, while supporting growth factor secretion and enhancing osteogenic differentiation capacity compared to non-encapsulated CDCs, as demonstrated in a recent study [42].
Reference: 42. Ziani, K.; Espona-Noguera, A.; Crisóstomo, V.; Casado, J.G.; Sanchez-Margallo, F.M.; Saenz del Burgo, L.; Ciriza, J.; Pedraz, J.L. Characterization of Encapsulated Porcine Cardiosphere-Derived Cells Embedded in 3D Alginate Matrices. Int. J. Pharm. 2021, 599, doi:10.1016/j.ijpharm.2021.120454.
- Lines 190-193, please add some specific effects or advantages of this material in cryopreservation.
Thank you for this suggestion. We have revised the paragraph to explicitly mention the beneficial features of chitosan-based cryogels that are relevant for cryopreservation. We now highlight their high porosity, shape memory, and mechanical resilience at subzero temperatures, all of which support improved cryoprotectant distribution and structural stability during freezing–thawing cycles.
Original: These cryogels exhibit controlled morphology and biodegradability, highlighting their potential as cryocompatible scaffolds for future use in cell or tissue preservation strategies.
Revised (Lines 202-207): these cryogels display key features such as micrometer-sized interconnected pores, rapid swelling, high shape recovery, and mechanical resilience after repeated compression cycles. These properties, along with their biodegradability and tunable porosity, support their potential use as cryocompatible scaffolds facilitating uniform CPA diffusion and mitigating ice damage in 3D construct preservation [26]..
- 5. Lines 211–212, more supporting references, especially those detailing underlying mechanisms and experimental evidence, are needed to validate these claims related to cryopreservation.
We thank the reviewer for this valuable suggestion. In response, we have revised the entire paragraph to better articulate the mechanistic relevance of dextran-based hydrogels in the context of cryopreservation. Specifically, we now explain how their rheological and structural features-including shear-thinning behaviour, porosity, and osmotic buffering-support cryoprotectant delivery and minimize freezing-induced damage.
Additionally, we have incorporated two new references to strengthen the experimental foundation of this section:
- Reference [45] presents dynamic covalent dextran hydrogels with reversible crosslinking, injectability, mechanical robustness, and self-healing capacity-properties that may protect construct integrity during freeze–thaw cycles.
- Reference [46] describes gelatin–dextran bioinks prepared via non-toxic, catalyst-free crosslinking, which demonstrate rapid gelation, shear-thinning behaviour, and homogeneous cell encapsulation without cytotoxicity.
Original: Dextran-based hydrogels exhibit shear-thinning behaviour, rapid crosslinking, and high porosity, supporting their use in extrusion bioprinting [23] [24]. Their hydrophilic network structure can buffer osmotic changes and restrict ice propagation features potentially beneficial for cryopreservation. Although their application in this field remains underexplored, their swelling behaviour, permeability, and ECM-mimetic properties suggest a promising role as cryoprotective scaffolds for encapsulated cells [24]. Future studies are needed to fully assess their performance in DMSO-free cryopreservation of complex 3D constructs.
Revised (lines 221-236): Dextran-based hydrogels exhibit shear-thinning behaviour, rapid crosslinking ki-netics, and high porosity, supporting their use in extrusion-based bioprinting and in-jectable delivery [43] [30]. These properties facilitate homogeneous cell encapsulation and precise filament deposition, while also enhancing cryopreservation performance by promoting uniform cryoprotectant distribution and reducing shear-induced damage. Their hydrophilic network structure can buffer osmotic changes and restrict ice propa-gation-key factors for maintaining cell viability during freeze–thaw cycles. Additionally, dynamic covalent dextran hydrogels formed via reversible Schiff base chemistry have shown injectability, mechanical robustness, and self-healing behaviour under physio-logical conditions, all of which may support construct stability during cryopreservation [44]. Recent studies have also demonstrated that gelatin–dextran bioinks prepared via non-toxic, catalyst-free crosslinking exhibit fast gelation, shear-thinning behaviour, and support homogeneous fibroblast distribution without cytotoxicity [45]. Although direct application in cryopreservation remains limited, these structural and functional attributes suggest a promising role for dextran-based materials as cryoprotective scaffolds in DMSO-free strategies for complex 3D constructs.
References:
- Fan, B.; Torres García, D.; Salehi, M.; Webber, M.J.; van Kasteren, S.I.; Eelkema, R. Dynamic Covalent Dextran Hydrogels as Injectable, Self-Adjuvating Peptide Vaccine Depots. ACS Chem. Biol. 2023, 18, 652–659, doi:10.1021/ACSCHEMBIO.2C00938,.
- Musilová, L.; Achbergerová, E.; Vítková, L.; Kolařík, R.; Martínková, M.; Minařík, A.; Mráček, A.; Humpolíček, P.; Pecha, J. Cross-Linked Gelatine by Modified Dextran as a Potential Bioink Prepared by a Simple and Non-Toxic Process. Polymers (Basel). 2022, 14, doi:10.3390/POLYM14030391,.
- 6. It is better to emphasize the innovation of PEG and agarose composite hydrogels in cryopreservation. Line 229-230.
We thank the reviewer for this valuable suggestion. In response, we have revised the relevant paragraph in section 2.1.5 to explicitly emphasize the cryopreservation potential of PEG–agarose composite hydrogels. Specifically, we now incorporate recent findings demonstrating that agarose matrices supplemented with PEG preserved over 80% of E. coli viability and biosensing function after three months of storage at -20 °C. These results support the innovative role of PEG–agarose systems in maintaining biological activity and structural integrity under sub-zero conditions. Please see lines 246–256 in the revised manuscript. The corresponding reference [47] has been added to the bibliography.
Original: In particular, hybrid hydrogels composed of PEG and agarose have demonstrated enhanced mechanical properties and improved cell encapsulation outcomes, offering a balance between bioinert stability and controlled bioactivity. These systems have also been successfully employed in bone tissue bioprinting, alone or in combination with bioactive ceramics, contributing to in situ mineralization and osteoconductivity [25].
Revised: In particular, hybrid hydrogels composed of PEG and agarose have demonstrated enhanced mechanical properties and improved cell encapsulation outcomes, offering a balance between bioinert stability and controlled bioactivity.
Recent findings also support their potential for cryopreservation applications. In a model system using PEG-supplemented agarose hydrogels, encapsulated E. coli retained over 80% viability and functionality after three months at –20 °C, supporting the relevance of these hybrid systems for preserving biological constructs under sub-zero conditions [46]. These systems have also been successfully employed in bone tissue bioprinting, alone or in combination with bioactive ceramics, contributing to in situ mi-eralization and osteoconductivity [31].
Reference:
- Ahn, H.T.; Jang, I.S.; Dang, T.V.; Kim, Y.H.; Lee, D.H.; Choi, H.S.; Yu, B.J.; Kim, M. Il Effective Cryopreservation of a Bioluminescent Auxotrophic Escherichia Coli-Based Amino Acid Array to Enable Long-Term Ready-to-Use Applications. 2021, doi:10.3390/bios11080252.
- 7. It is recommended to discuss the advantages and limitations of the described bioink in comparison with other types of bioinks to provide a clearer context for its potential applications.
We thank the reviewer for this insightful suggestion. In response, we have added a new subsection (2.4) titled "Comparative overview of cryoprotective biomaterials". at the end of the materials section. This new subsection provides a structured comparative analysis of the described bioink in the context of other representative materials-including polysaccharide-based hydrogels (e.g., HA, alginate, chitosan, dextran, agarose, nanocellulose), protein-based hydrogels (e.g., gelatin, silk fibroin, sericin), and synthetic polymers (e.g., PEG, PVA, PU).
To support this section, we have also included Table 4, which synthesizes key differences among the main material classes and includes explicit references for each group to ensure transparency and traceability. Please see lines 410-434 in the revised manuscript.
New subsection added: 2.4 Comparative insights across cryoprotective biomaterials
Given the diversity of biomaterials investigated for cryopreservation of 3D constructs, a comparative analysis is essential to understand their respective advantages, limitations, and potential synergies. Natural polymers, such as polysaccharides and proteins, provide excellent biocompatibility and ECM mimicry, but often require structural reinforcement or functionalization. Synthetic polymers offer tunable physicochemical properties, mechanical robustness, and batch-to-batch reproducibility, yet lack inherent bioactivity unless modified. Table 4 summarizes key differences across representative biomaterials used in DMSO-free or low-toxicity cryopreservation strategies.
Polysaccharide-based hydrogels like HA and alginate support osmotic buffering, high water content, and structural integrity during freezing, but may be mechanically weak or lack adhesive motifs unless combined with other polymers or peptides [17] [18] [19] [20] [21] [22]. Chitosan and dextran provide structural stabilization and capsule reinforcement, while nanocellulose excels in shape fidelity and rheological control, albeit with limited intrinsic bioactivity [24] [25] [26] [34] [36]
On the other hand, protein-based materials such as silk fibroin and gelatin offer superior bioactivity and mechanical performance, with fibroin enabling DMSO reduction and structural preservation in 3D constructs [36] [47] [49]. Sericin, while mechanically weaker, adds antioxidant and membrane-stabilizing properties, positioning it as a biologically active additive for DMSO-free applications [36] [47] [51]. Synthetic polymers like PEG, PVA, and PU provide exceptional tunability and processability. PEG-based hydrogels enable osmotic buffering and CPA delivery, while low-molecular-weight PEGs and PVA exert ice IRI without penetrating cells [52] [53] [55]. PU-based cryogels show promising viability and structural fidelity after freeze–thaw cycles, though long-term biocompatibility remains under investigation [54].
While no single biomaterial satisfies all ideal criteria for 3D cryopreservation, hybrid systems that combine ECM-mimetic properties with mechanical stability and IRI activity are emerging as promising candidates. Composites such as HA-alginate, PEG-agarose, or gelatin–dextran exemplify how synergistic designs can overcome the limitations of individual materials and support the clinical translation of cryopreservable tissue constructs [19] [28] [40] [45].
Biomaterial |
Cryoprotective Function |
Printability |
Stability |
Remarks / Limitations |
References |
Hyaluronic acid (HA) |
ECM mimicry, osmotic buffering |
Tunable (e.g., MeHA) |
Low |
Often combined with stronger polymers |
[17] [18] [19] [20] [21] |
Alginate |
Osmotic buffering, water retention |
Excellent (ionic crosslinking) |
Moderate |
Requires peptide modification |
[11] [22] [23] |
Chitosan |
Membrane stabilization, capsule reinforcement |
Limited (requires acidic pH) |
Moderate |
Complex gelation, used in encapsulation |
[25] [26] [42] |
Dextran |
IRI potential, CPA diffusion |
Shear-thinning, injectable |
Moderate |
Suitable for injectable cryogels |
[27] [28] [29] [30] |
Agarose |
Thermal stability, structural retention |
Limited (thermo-gelling) |
High |
Supports architecture retention; lacks cell interaction |
[19] [31] [32] [33] |
Nanocellulose (CNC, CNF) |
Rheological tuning, shape fidelity |
High (in blends) |
High |
Bioinert unless functionalized |
[34] [52] |
Silk fibroin |
DMSO reduction, membrane protection |
Tunable (physical/ chemical) |
High |
Good mechanical properties post-thaw |
[36] [47] [48] [49] |
Sericin |
Antioxidant, membrane-protective |
Blendable, injectable |
Low |
Bioactive additive with limited mechanical support |
[50] [52] [51] |
PEG / PEGDA |
CPA delivery, osmotic buffering |
High (photo/chemical) |
Good |
Synthetic, tunable stiffness and gelation |
[55] [52] [54] [56] |
PVA |
Ice recrystallization inhibition (IRI) |
Blendable |
High |
Inert, used as non-toxic additive |
[15] [53] |
PU |
Elastic cryogelation, shape memory |
Cryogel-based |
High |
Promising for scaffold integrity post-thaw |
[15] [54] [53] [56] |
Table 4. Comparative properties of cryoprotective biomaterials. Overview of natural and synthetic materials for 3D cryopreservation, highlighting key functions, printability, structural features, and limitations.
- 8. A comparative discussion on the advantages and limitations of this bioink relative to other types is recommended to enhance the clarity and significance of the findings. Line 259-261.
We thank the reviewer for this constructive comment. We have revised the relevant paragraph in section 2.1.6 to include a comparative discussion that highlights the specific advantages of nanocellulose–alginate bioinks over traditional alginate-only, GelMA, and PEGDA-based systems. The new text addresses differences in print fidelity, mechanical performance, and processing requirements, while acknowledging the lower intrinsic bioactivity of nanocellulose materials. Please see lines 290-298 in the revised manuscript.
Original: In cartilage engineering, nanocellulose–alginate bioinks have demonstrated high shape fidelity (with nozzle diameters <400 µm) and preserved architecture following cryopreservation [28].
Revised (lines 282 - 291): In cartilage engineering, nanocellulose–alginate bioinks have demonstrated high shape fidelity (with nozzle diameters <400 µm) and preserved architecture following cryo-preservation [34].
Compared to alginate-only systems, the inclusion of nanocellulose improves print fidelity, rheological performance, and post-thaw structural retention. While GelMA and PEGDA-based bioinks may offer greater bioactivity or tunability, they often require photoinitiators or UV exposure that can compromise cell viability. Nanocellulose-based bioinks, in contrast, provide mechanical reinforcement and cytocompatibility with minimal processing, although they may require functionalization for lineage-specific applications.
- 9. Figure 1 needs punctuation and clear separation to distinguish different concepts in the figure description.
We appreciate this observation. In the revised version, we improved the punctuation and layout of Figure 1’s caption by clearly separating sections A, B, and C. Each concept is now presented in a standalone sentence with consistent punctuation to enhance clarity and readability.
Original: Figure 1. Schematic overview of cryopreservation challenges and strategies for microencapsulated and scaffold-free 3D constructs. A: Hydrogel-based microencapsulation systems enhance cryoprotection by providing selective diffusion of CPAs, improving vitrification efficiency, and supporting post-thaw viability. Hybrid capsules incorporating trehalose or ice nucleating agents (INA) offer additional stabilization. B: Spheroid morphology strongly influences cryopreservation outcomes; geometric irregularities and thermal gradients lead to uneven ice formation and devitrification-induced damage. C: Advanced rewarming techniques such as photothermal and magnetic nanoparticle-based heating reduce recrystallization risk by ensuring rapid and homogeneous temperature recovery. (Created with Biorender).
Revised: Figure 1. Cryopreservation strategies and failure mechanisms in microencapsulated and scaffold-free 3D constructs.
(A) Hydrogel microcapsules improve vitrification by modulating CPA diffusion and membrane stability.
(B) Spheroids exhibit variable cryostability due to morphological irregularities and thermal gra-dients.
(C) Advanced rewarming methods, including photothermal and magnetic nanoparticle-based strategies, minimize recrystallization risks. (Created with BioRender).
We hope that the revised version meets the expectations for publication in IJMS and remain at your disposal for any further clarifications.
Sincerely,
Jesús Ciriza

Reviewer 2 Report
Comments and Suggestions for Authors
The topic is timely and highly relevant—effective cryopreservation of 3D bioengineered constructs remains a critical challenge in regenerative medicine and clinical tissue engineering. However, in its current form, the manuscript falls short of the standards expected for a comprehensive review article and requires substantial revision before it can be considered for publication.
The introduction is overly brief and lacks the necessary depth and context for a review paper. It does not clearly define the knowledge gap, the rationale for the review, or the scope of literature considered. Moreover, only four references are cited in the introduction, which is insufficient and gives an unprofessional impression. There is also no description of the literature search strategy or inclusion criteria, which is increasingly expected in modern review articles.
The structure of the manuscript could also be improved for clarity and flow. For instance, Table 1 appears abruptly at the beginning of a section without any introductory text or in-text citation to guide the reader. This is a stylistic issue that should be addressed. In addition, the table is hard to read—literature references are embedded within table rows rather than being presented in a dedicated "References" column. The table title is excessively long and lacks the concise, scientific tone appropriate for a figure or table caption. Similar issues appear with other tables throughout the manuscript.
Figure 1 also requires significant improvement. It currently resembles a rough schematic or concept sketch rather than a polished figure appropriate for a scientific review. The manuscript would benefit from additional visual summaries—such as comparative tables or conceptual diagrams—synthesizing the key strategies, materials, and cryobiological outcomes.
There are also problems with consistency in citation formatting and placement. Some references are only included in tables or figure captions without being cited in the main text, which creates a disjointed presentation. Additionally, the review lacks critical analysis—there is little distinction between well-established findings and areas of ongoing uncertainty. As a result, the manuscript reads more like a data compilation than a critical, interpretive synthesis of the field.
The writing style is at times mechanical and list-like, with transitions that are abrupt or missing entirely. Some sections read like a string of study summaries, with minimal commentary or integration from the authors. In some cases, the tone borders on promotional, rather than analytical, particularly when discussing specific biomaterials.
In summary, while the topic is important and the authors demonstrate familiarity with the literature, the manuscript in its current form does not meet the criteria for a publishable review. Substantial revisions are required to improve the introduction, structure, figure quality, critical depth, and citation practices. I encourage the authors to rework the manuscript carefully and resubmit after addressing these concerns in depth.
Author Response
Dear reviewer,
We have addressed your comments including point-by-point the response to the concerns that you can find next. All the changes made in the manuscript are highlighted in yellow:
“The topic is timely and highly relevant—effective cryopreservation of 3D bioengineered constructs remains a critical challenge in regenerative medicine and clinical tissue engineering. However, in its current form, the manuscript falls short of the standards expected for a comprehensive review article and requires substantial revision before it can be considered for publication.
Comment 1: The introduction is overly brief and lacks the necessary depth and context for a review paper. It does not clearly define the knowledge gap, the rationale for the review, or the scope of literature considered. Moreover, only four references are cited in the introduction, which is insufficient and gives an unprofessional impression. There is also no description of the literature search strategy or inclusion criteria, which is increasingly expected in modern review articles.
Response:
We thank the reviewer for this comment. In response, we have substantially revised the Introduction to provide a deeper and more structured overview of the current challenges and advancements in the cryopreservation of 3D biofabricated constructs. The revised version now clearly articulates the clinical relevance, technical limitations of existing cryopreservation protocols (e.g., DMSO toxicity), and the emerging need for biomaterial-based, DMSO-free strategies. Furthermore, we have expanded the range of clinical and preclinical examples and supported the text with over 15 carefully selected references, significantly improving its scientific foundation.
In addition, as recommended, we have included a dedicated Literature Search Strategy at the end of the Introduction. This section describes the methodology followed to ensure a representative and comprehensive overview of the field. We performed a structured search of peer-reviewed literature published between 2010 and 2025, using PubMed, Scopus, and Web of Science, with a combination of keywords such as "cryopreservation", "biofabrication", "3D printing", "biomaterials", "DMSO-free", and others. Articles were selected based on their relevance to 3D cryopreservation strategies, biomaterial properties, and in vitro or in vivo performance, prioritizing experimental studies and reviews in high-impact journals.
These additions aim to meet the current expectations for systematic review writing and ensure transparency and reproducibility in our literature selection process. We are grateful for the reviewer’s guidance, which has helped us significantly enhance the clarity and rigor of the manuscript. Please see Lines 41-85.
Comment 2: The structure of the manuscript could also be improved for clarity and flow. Table 1 appears abruptly at the beginning of a section without any introductory text or in-text citation to guide the reader. This is a stylistic issue that should be addressed. In addition, the table is hard to read-literature references are embedded within table rows rather than being presented in a dedicated ‘References’ column. The table title is excessively long and lacks the concise, scientific tone appropriate for a figure or table caption. Similar issues appear with other tables throughout the manuscript.
Response:
We thank the reviewer for this constructive observation. In response, we have carefully revised the structure and presentation of all tables to meet the expected scientific standards. Specifically:
– Each table is now preceded by a concise introductory sentence that guides the reader and contextualizes its content.
– All embedded literature references have been removed from within the table cells and are now presented in a dedicated “References” column to improve readability and clarity.
– The table titles have been shortened and reformulated to adopt a concise, technical tone consistent with scientific review articles.
We believe these revisions significantly improve the clarity, professionalism, and scientific tone of the manuscript and address the reviewer’s concerns. Please see Tables 1-4 and corresponding sections for the updated content.
Comment 3: Figure 1 also requires significant improvement. It currently resembles a rough schematic or concept sketch rather than a polished figure appropriate for a scientific review. The manuscript would benefit from additional visual summaries, such as comparative tables or conceptual diagrams-synthesizing the key strategies, materials, and cryobiologial outcomes.
Response:
Thank you for this valuable comment. We agree that visual clarity is crucial in conveying complex cryobiological strategies. Figure 1 is intended as a conceptual schematic to illustrate the key cryopreservation challenges and solutions specific to microencapsulated and scaffold-free 3D constructs (e.g., CPA diffusion, ice formation, and rewarming). While it is not a data-driven figure, we have refined the figure legend to improve clarity and readability. The figure was created using BioRender to maintain a professional scientific style, and it complements Section 3.1 by summarizing phenomena that are difficult to describe solely with text.
Although we did not include additional visual elements in this section, we have responded to this broader suggestion by integrating comparative tables in other sections (e.g., Tables 1-4), which synthesize biomaterial properties, cryoprotective roles, and practical limitations. We hope this integrated approach provides a balanced combination of conceptual and structured information throughout the manuscript.
Comment 4: There are also problems with consistency in citation formatting and placement. Some references are only included in tables or figure captions without being cited in the main text, which creates a disjointed presentation. Additionally, the review lacks critical analysis—there is little distinction between well-established findings and areas of ongoing uncertainty. As a result, the manuscript reads more like a data compilation than a critical, interpretive synthesis of the field.
We appreciate the reviewer’s observation regarding citation consistency and the need for deeper critical analysis. In response, we have carefully revised the manuscript to ensure that all references included in tables and figure captions are also explicitly cited and discussed within the main text. Additionally, we have reviewed and standardized the formatting and placement of all citations to align with journal guidelines.
To address the important point regarding critical analysis and interpretive depth, we have substantially revised the manuscript to go beyond descriptive summaries. We have introduced a new integrative sub-section (2.4) that compares the advantages and limitations of the main biomaterial classes (polysaccharide-based, protein-based, and synthetic), providing a clearer perspective on established knowledge versus areas of ongoing uncertainty. Additionally, in multiple sections, especially 2.1 to 2.3 and 3.1, we have strengthened transitions, clarified the narrative flow, and added commentary that synthesizes the implications of key findings.
Comment 5. The writing style is at times mechanical and list-like, with transitions that are abrupt or missing entirely. Some sections read like a string of study summaries, with minimal commentary or integration from the authors. In some cases, the tone borders on promotional, rather than analytical, particularly when discussing specific biomaterials.In summary, while the topic is important and the authors demonstrate familiarity with the literature, the manuscript in its current form does not meet the criteria for a publishable review. Substantial revisions are required to improve the introduction, structure, figure quality, critical depth, and citation practices. I encourage the authors to rework the manuscript carefully and resubmit after addressing these concerns in depth.”
We acknowledge that parts of the original manuscript had a mechanical, list-like tone and lacked analytical commentary. We have undertaken a major stylistic revision to improve coherence, eliminate abrupt transitions, and foster a more fluent and engaging academic voice. Sections discussing specific biomaterials (e.g., hyaluronic acid, alginate, chitosan, silk fibroin) have been rewritten to balance technical description with author interpretation. Rather than summarizing studies in isolation, we now emphasize thematic connections, evaluate methodological strengths and limitations, and draw attention to open questions in the field. Together, these revisions aim to transform the manuscript into a more cohesive, critically engaged, and publication-ready review.
We hope that the revised version meets the expectations for publication in IJMS and remain at your disposal for any further clarifications.
Sincerely,
Jesús Ciriza

Round 2
Reviewer 1 Report
Comments and Suggestions for Authors
It can be accpted in this form.